# Effects of Heating Treatment on the Physicochemical and Volatile Flavor Properties of Argentinian Shortfin Squid *(Illex argentinus*)

**DOI:** 10.3390/foods13071025

**Published:** 2024-03-27

**Authors:** Jiagen Li, Zhaoqi Li, Shanggui Deng, Soottawat Benjakul, Bin Zhang, Jiancong Huo

**Affiliations:** 1Key Laboratory of Health Risk Factors for Seafood of Zhejiang Province, College of Food Science and Pharmacy, Zhejiang Ocean University, Zhoushan 316022, China; jiagen10161828@outlook.com (J.L.); lizhaoqi1213163@163.com (Z.L.); dengshanggui@163.com (S.D.); zhangbin@zjou.edu.cn (B.Z.); 2International Center of Excellence in Seafood Science and Innovation, Faculty of Agro-Industry, Prince of Songkla University, Songkhla 90112, Thailand; soottawat.b@psu.ac.th

**Keywords:** squid, heating temperature, physicochemical properties, volatile flavor properties

## Abstract

In this study, the effect of different heating temperatures (80, 90, 100, and 121 °C) on the physicochemical and volatile flavor properties of fried mantles (Argentinian shortfin) was investigated. The squid mantles were soaked in a maltose syrup solution (20% *w*/*v*) for 10 s and fried in soybean oil for 10 s (160 °C), vacuum-packed, and processed at different temperatures for 10 min. Then, the squid mantles were subjected to colorimetric analysis, sensory evaluation, free amino acid analysis, and texture profile analysis. In addition, the volatile organic compounds (VOCs) in the squid mantles were analyzed. The results revealed that lower treating temperatures (80 and 90 °C) improved the chromatic and textural properties, along with organoleptic perception. Additionally, the content of amino acid in the squid mantles treated at 121 °C was significantly lower than that of the samples treated at other temperatures (*p* < 0.05). Headspace-gas chromatography-ion mobility spectrometry (HS-GC-IMS) was used to detect 41 VOCs, including their monomers and dimers. Among these detected VOCs, the contents of alcohols, ketones, and pyrazines were positively correlated with temperature. However, the content of aldehydes in the squid mantles gradually decreased as the heating temperature increased (*p* < 0.05). The combined HS-GC-IMS and E-nose results revealed that the lower temperatures (80 and 90 °C) were more suitable for flavor development and practical processing. This study provides valuable information for properly controlling the heating process of squid products, as well as flavor and practical applications for the aquatic industry.

## 1. Introduction

Squid is widely regarded as a high-economic-value seafood and is mainly consumed in East Asian countries [1]. The Argentinian shortfin squid (*Illex argentinus*) is one of the most important species and is popularly consumed owing to its nutritional advantages, such as its high content of protein and taurine and its low content of fat [2]. Heating is a common and effective method of preserving squid products and extending their shelf life. The heating method inhibits the growth/reproduction of microorganisms, inactivates enzymes, and endows squid products with a special flavor via physicochemical reactions. However, the chemical composition of squid muscle (including the rich nutrients, high content of water, and various precursor substances), which greatly affects the final commercial quality of squid products, is highly susceptible to the action of heating [3]. Some undesired heat-induced quality changes, including textural deterioration, discoloration, nutrient loss, and flavor deterioration, inevitably occur due to high temperatures or long-term heating during traditional thermal treatments.

Most aquatic food products are highly valued for their nutritional content and sensory appeal, especially flavor [4]. Typical flavor compounds with low thresholds (such as aldehydes), which are mostly derived from lipolysis or oxidation, are responsible for the final flavor quality of the products [5,6,7]. In addition to the concentrations of the flavor compounds, the interactions between the flavor compounds and other food components (including protein and fat) contribute to the intensity of the flavor perception [5]. Furthermore, studies have shown that the Maillard reaction, lipid oxidation, and protein degradation play a significant role in generating critical flavor compounds [8]. The intensity of the heat-induced chemical reactions and the concentration of flavor compounds derived from the above-mentioned chemical reactions greatly depend on the temperature and cooking time [9]. Aquatic product muscle is severely contracted under high temperatures due to its unique structure of muscle protein, resulting in the textural deterioration (extra hardness) of the muscle [10]. Additionally, improper cooking temperatures might lead to excessive chromatic deterioration and produce harmful heterocyclic compounds, negatively affecting the final product [11]. Thus, the commonly recommended cooking conditions for aquatic food processing are approximately 80–100 °C and 10–20 min; for example, the recommended conditions for surimi products and aquatic sauces are 90 °C and 10 min; for crab meat, 90 °C and 10–15 min; for fish mince product, 80 °C and 20 min; and for freshwater prawn, 65 °C and 10 min [12,13,14].

Fried squid is a typical Chinese-style ready-to-eat squid product. Owing to rapid economic development and the quest for convenient aquatic food, the demand for ready-to-eat squid products has rapidly increased with a broader market. However, current reports on the flavor of squid products mainly focus on the optimization/improvement in the traditional processing methods and on the effect of adding ingredients during processing [15,16,17,18,19,20]. There are few reports on the impact of temperature on the physicochemical and volatile flavor properties of squid products [15,21]. Therefore, this study aimed to (1) investigate the effect of temperature on the nutritional and physicochemical properties of squid products; (2) establish and characterize the flavor fingerprints of squid treated at different temperatures; and (3) provide useful information on the optimal heating temperature for squid.

## 2. Materials and Methods

### 2.1. Chemicals

Methanol was purchased from Merck Drugs & Biotechnology Ltd. (Beijing, China). HPLC-grade acetonitrile was obtained from Fisher Companies, Inc. (Fair Lawn, NJ, USA). Amino acid standards were purchased from Wako Pure Chemical Industries, Ltd. (Osaka, Japan). The filter (0.22 μm), trichloroacetic acid (TCA), HCl, sodium citrate, and other chemicals obtained from Seebio Biotech (Shanghai) Co., Ltd. (Shanghai, China) were of analytical grade.

### 2.2. Squid Samples and Treatments

Fresh Argentinian shortfin squid (*Illex argentinus*, average weight 400 ± 10 g) was purchased from a local aquatic products market (Laoqi Market, Zhoushan, China). The obtained squid samples were refrigerated with flake ice in a polystyrene box and immediately transported to the laboratory within 20 min. Before the processing of squids, they were eviscerated and separated into the head, wrist, and mantle using a knife after washing. The mantle had an average body scale of 3 cm × 3 cm × 0.5 cm and a weight of 10 ± 2 g. All squid mantles were soaked in maltose syrup solution (20% *w*/*v*) for 10 s and then fried in soybean oil for 10 s (160 °C). After the frying process, the squid mantles were vacuum-packed with nylon/polyethylene bags and heated at 80, 90, 100, and 121 °C in a thermostat water bath (HH-2, Lichen Co., Ltd., Shanghai, China) and high-pressure sterilizer (DGL-35B, Lichen Co., Ltd., Shanghai, China) for 10 min for further analysis. The squid mantles, which were treated similarly to other samples but without heating treatment, were considered the unheated samples (UH).

### 2.3. Colorimetric Analysis

Changes in Hunter’s color were assessed by measuring the *L**, *a**, and *b** values (brightness, *L**; red–green index, *a**; and yellow–blue index, *b**) using a color meter (CR-10 Analyzer, Konica Minolta, Tokyo, Japan) [22]. The white standard board was used for calibration.

### 2.4. Sensory Evaluation

Sensory evaluation was conducted following the method of Paarup et al. [6]. For the descriptive analysis of squid products, a panel of 10 trained assessors (aged 22–28) participated in the sensory evaluation. Furthermore, they were asked to avoid cooking and using household cleaners one hour before the test and to follow the general behavior rules preceding sensory evaluations (e.g., not to smoke, eat, or drink, except for water, prior to the test). The parameters, including the odor, taste, tenderness, and juiciness of the squid mantles, were awarded points on a scale of 1–7, with 7 expressing optimum quality and 1 the rejection limit. Total scores were calculated according to the results mentioned above.

### 2.5. Free Amino Acid Analysis

The free amino acid contents of the squid mantles were determined as previously described by Oh et al., with slight modifications [23]. Minced squid mantle (1.0 g) was homogenized with 20 mL of deionized water, and 10 mL of 5% (*w*/*v*) TCA solution was added to the homogenate. After the homogenate was incubated at 4 °C for 1 h, the mixture was centrifuged at 10,000× *g* for 15 min at 4 °C (CR22G, Hitachi Corp., Tokyo, Japan). Then, the supernatant was collected and filtered through a 0.22 μm filter. The content of free amino acids in the supernatant was analyzed using the automatic amino acid analyzer (L-8900, Hitachi Corp., Tokyo, Japan) and expressed as mg/g muscle. The taste activity value (TAV) was calculated using the following formula:$$TAV=\frac{C(mg/g)}{Threshold(mg/g)}$$
where C is the concentration of an individual compound (mg/g) and Threshold is its corresponding taste threshold (mg/g).

### 2.6. Texture Profile Analysis

The texture of the squid mantles was analyzed using a texture analyzer (TA.XT PlusC, Stable Micro Systems, Godalming, UK). Texture profile analysis of the squid mantles was performed following the method described by Zhou et al. [24]. The specific parameters used were as follows: a pretest speed of 1.0 mm/s, a test speed of 5.0 mm/s, a post-test speed of 5.0 mm/s, a holding time of 3 s, and a compression ratio of 30%. The texture characteristics were determined by analyzing the force–time curves obtained from the analysis.

### 2.7. Volatile Organic Compound Analysis

Headspace-gas chromatography-ion mobility spectrometry (HS-GC-IMS) was employed for volatile organic compound analysis according to the method described by Wang et al., with slight modifications, using a FlavourSpec^®^ GC-IMS instrument (Gasellschaft für Analytische Sensorsysteme, Dortmund, Germany) equipped with an MXT-WAX chromatographic column (30 m in length and 0.53 mm in internal diameter) using N_2_ as a carrier gas [25]. Precisely, 10 g of squid mantle was homogenized in a blender for 5 min. Then, 2 g of mashed squid mantle was transferred to a 20 mL headspace vial and incubated for 15 min at 60 °C. The testing procedures were conducted under the following conditions: injection volume of 500 μL, injection temperature of 85 °C, and incubation rotation rate of 500 rpm. The gas chromatography conditions were as follows: column temperature of 60 °C, analysis time of 20 min, and initial carrier flow rate of 2 mL/min for 2 min. Then, the carrier flow rate increased to 10 mL/min for 8 min and finally increased to 100 mL/min for 10 min.

### 2.8. Electronic Nose Analysis

Electronic nose (E-nose) analysis was performed using a PEN3 E-nose (PEN3, Airsense, Schwerin, Germany) according to the method described by Li et al. [26]. The detection was performed with 10 different metal oxide gas sensors. Precisely, 3 g of squid mantle was placed into a 20 mL headspace vial, sealed, and equilibrated in a 60 °C water bath for 20 min. The flavor was detected via a probe, which was inserted into the sealed vial to exude the aroma of the chemical substance through a separative membrane. The measurement time and flow rate were 90 s and 400 mL/min, respectively. The signals were selected and analyzed using WinMuster software.

### 2.9. Statistical Analysis

All experiments were triplicated. The HS-GC-IMS data were qualitatively analyzed using the HS-GC-IMS library search and laboratory analysis viewer software (FlavourSpec^®^, G.A.S., Dortmund, Germany). The sensory analysis was conducted using SPSS 13.0 software. Each result was expressed as a mean ± standard deviation, and *p* < 0.05 indicated statistically significant differences using Duncan’s multiple range tests.

## 3. Results and Discussion

### 3.1. Colorimetric Analysis

According to the results of Figure 1A, the Hunter’s *L** values of the squid mantles treated with different temperatures were 42.8 (80 °C), 41.6 (90 °C), 36.4 (100 °C), and 17.8 (121 °C), respectively. No significant differences were observed in the *L** values of the squid mantles treated with 80 and 90 °C (*p* > 0.05), but significant differences were observed in the *L** values of the squid mantles treated at 121 °C (*p* < 0.05). In this study, the *L** value of the squid mantles significantly decreased after 10 min of heating treatment at 121 °C. After the treatment, the squid mantle skins exhibited browning, resulting in a great loss in commercial value. This phenomenon might be attributed to the Maillard reaction, an important chemical reaction that affects the quality of food under extreme conditions, such as long-time heating at high temperatures [27]. Furthermore, significant increases in the values of *a** and *b** in all the squid mantle samples were observed, and the value of *b** of the squid mantles (121 °C) was significantly higher than that of the other samples (*p* < 0.05), which contributed to the browning of the squid mantles [28]. These color changes were further confirmed by photographic records (Figure 1B).

Appearance and pleasantness play a vital role in product acceptability and preference, especially for aquatic products. Aquatic muscles tend to discolor after heating treatment, thus significantly decreasing their market value. The results shown in Figure 1A indicate that high temperatures greatly affected the chromatic characteristics of the squid mantles through the activation of chemical reactions (especially the Maillard reaction) [29]. This process produces a large amount of pigment by generating melanoidin, which results from 5-hedroxymethylfurfural, the essential product of the advanced stage of the Maillard reaction, and further reacts with primary amine groups [30]. Geng et al. also found out that the Maillard reaction mainly generated pigments, especially in a reaction environment with numerous carbonyl and amino compounds under high temperatures [31]. In this study, the necessary conditions for the initiation of the Maillard reaction were met, consequently leading to changes in the chromatic properties (*L**, *a**, *b**) of the squid mantles. Similar findings were proposed by Ning et al., who confirmed that high treating temperatures (higher than 150 °C) decreased the *L** value and increased the *a** and *b** values of squid [15]. In addition, Wang et al. reported that heating temperature effectively affected the intensity of the Maillard reaction through the participation of water molecules [32]. This result also shows the importance of controlling the heating temperature for aquatic foodstuffs, which is characteristic of high water content.

### 3.2. Sensory Evaluation

The results of the organoleptic perception evaluated by the panels are presented in Table 1. The squid mantles treated with 121 °C exhibited the lowest score (2.81) among the four samples. This result was consistent with the results of the colorimetric analysis. Additionally, nine panelists detected scorched off-odors in the sample treated at 121 °C, indicating the probably slight charking in protein caused by the high-temperature treatment. Higher sensory scores were observed in the lower-temperature-treated samples, especially for the squid mantles treated at 80 and 90 °C. As shown in Table 1, no significant differences were observed in all the evaluation parameters between the squid mantles treated at 80 and 90 °C (*p* > 0.05). These two samples exhibited typical squid taste and flavor, which possibly resulted from the ketones and esters produced through the Maillard reaction or fat oxidation [33,34]. Furthermore, the tenderness values of these two samples (treated at 80 and 90 °C) exhibited higher scores compared with the other two samples (treated at 100 and 121 °C) (*p* < 0.05). This result was obtained because the squid muscle fibers were present in a radial and circular orientation and tightly interposed with connective tissue fibers. The squid muscle became sticky and tough after heating at a temperature above 100 °C, thus resulting in strong hardness [35,36]. Afsin and Nalan also indicated that the lower temperature (below 80 °C) was helpful in maintaining the sensory quality of squid, which was processed by sous-vide cooking [21]. The results of the sensory evaluation revealed that the temperature significantly influenced the sensory quality of the squid mantles. A temperature of 121 °C, which is frequently applied to the sterilization of animal meat products, is probably not suitable for processing squid or its products with the abundant precursor ingredients of the Maillard reaction unless the heating time is strictly restricted.

### 3.3. Free Amino Acid Analysis

The compositions and contents of amino acids in all the samples are shown in Table 2. As shown in Table 2, the amino acid content in the UH samples and the four heating-treated samples was rich in glutamic acid, alanine, arginine, and proline. However, the heating treatment at the lower temperatures (80 and 90 °C) had no significant effect on the amino acid composition or content of the squid mantle muscle protein (*p* > 0.05). Similar findings were observed by Charlotte et al., who revealed that a lower heating temperature (below 100 °C) was helpful in maintaining the original amino acid components of squid muscle protein [19]. Conversely, heating treatment at high temperatures (above 100 °C) significantly reduced the total content of amino acids. The total amino acid contents of the samples (100 °C, 683.10 mg/g; and 121 °C, 521.27 mg/g) were significantly lower than those of the UH sample (762.17 mg/g). The samples treated with 80 °C (743.67 mg/g) and 90 °C (737.27 mg/g) (*p* < 0.05) showed a negative correlation between temperature and total amino acid content. Additionally, taste activity values (TAVs) are frequently used to evaluate the contribution of free amino acids to the taste of food [37]. According to the results in Table 2, the amino acids with a TAV value higher than 1 in all the samples were glutamic acid, alanine, and arginine. The content of glutamic acid in the 121 °C treated sample was 143.33 mg/g (approximately 50% content in relation to the sample treated with 90 °C). This result might explain the reason for the lowest sensory evaluation score in the 121 °C treated sample. Meanwhile, the arginine content of the 121 °C treated sample was 58.00 mg/g. Arginine has been regarded as a negative amino acid (a substance with bitterness) for the taste of food [38]. However, according to the results of Shabbir et al. and Huang et al., the taste of arginine can be masked using amino acids with a sweet taste and delicate flavor, thus effectively restricting the negative effect of arginine on the sensory properties of squid [39,40].

### 3.4. Texture Analysis

The effects of the heating treatment on the springiness and chewiness of the squid mantles were evaluated (Figure 2). Significant differences in springiness values between the samples treated with 100 and121 °C and the other two samples (80 and 90 °C) (*p* < 0.05) were observed. However, no significant differences in springiness values between the samples treated at 80 and 90 °C (*p* > 0.05) were observed. Additionally, all samples showed a similar trend in chewiness and springiness values. The chewiness values of the four samples gradually decreased with the increase in heating temperature. The samples treated at 80 and 90 °C exhibited higher chewiness values (102.2 and 98.6 N·mm) compared with the samples treated at 100 °C (59.6 N·mm) and at 121 °C (45.8 N·mm) (*p* < 0.05). These results suggested that the high heating temperatures (100 and 121 °C) were effective in reducing the springiness and chewiness of the squid mantles. Thus, the sensory evaluation results regarding tenderness were associated with the changes in the springiness and chewiness values of the squid mantles. In squid muscle, myofibrillar protein (MP) stability and connective tissue strength play important roles in muscle texture. In this study, the springiness and chewiness values of the squid mantles significantly decreased, mainly due to the contraction of MP resulting from the heating treatment. Barekat and Soltanizadeh proposed a similar idea and also observed a considerable textural deterioration in squid muscle after improper heating treatment, resulting in increased hardness and decreased springiness and chewiness [41]. Furthermore, Shi et al. found that myosin (the main component of the MP matrix) was critical to the textural properties of muscle and was easy to deform and aggregate during heating [42]. Thus, undesired heat-induced textural changes (extra hardness) occurred. In this study, the results clearly demonstrated that heating temperature significantly affected the textural properties of the squid mantles.

### 3.5. Analysis of HS-GC-IMS

#### 3.5.1. Topographic Plots

The volatile compounds of the samples are represented by spots on 2D topographic plots (Figure 3A). The red vertical line of the spectrum signifies the reactive ion peak (RIP), which was applied to detect the activity of the detected substances [43]. The volatile organic compounds (VOCs) of the squid mantles, which are primarily distributed on the right side of the RIP, were considered active and analyzed. The red spots with varied brightness levels and sizes represent the different concentrations of each substance. The *x*-axis, *y*-axis, and *z*-axis of the 3D topographic plot represent the separated drift time of the ions, the detected retention time of the substances, and the signal intensity of the ionic compounds, respectively (Figure 3B). It can be seen that most VOCs of the squid mantles appear in a drift time range of 6.0–8.0 ms and a retention time within 1000 s (Figure 3A,B). Moreover, it can be seen that the distributions of the ion peaks of the UH samples and the heating-treated squid mantles are extremely similar, whereas the abundance of the compounds in the five samples differs according to the heights of the detected peak signals. Furthermore, as the temperature increased, the contents of VOCs in the heating-treated squid mantles, especially those treated at 121 °C, significantly changed, as shown by the height and brightness of the spots. This change indicates that temperature significantly influences the amount of VOCs. Thermal-sensitive chemical compounds such as trimethylamine, a derivative of trimethylamine oxide (TMAO), decomposed during the heating process and generated various products with low molecules, which consequently contributed to their abundance in the squid mantles treated at 121 °C [44].

#### 3.5.2. Fingerprints

The method of fingerprints was effective in distinguishing substances with close relation [45]. As shown in Figure 3C, 41 VOCs, including their monomers (abbreviated as M) and dimers (abbreviated as D), and other 18 unidentified compounds, were detected using the HS-GC-IMS method. Significant differences in the VOC profiles were observed between the samples treated with different temperatures. The total amount of ketones, pyrazines, and trimethylamine in the high-temperature-treated group (121 °C) was significantly higher than that in the other groups (*p* < 0.05). This result was consistent with that of Zhu et al., who pointed out that high temperatures contribute to the degradation of lipids and proteins, resulting in the formation of ketones and trimethylamine [44]. Additionally, chemical substances with high concentrations, including 1-hydroxy-2-propanone-M/D, 3-hydroxybutan-2-one-M/D, 2-methylpyrazine-M/D, 2-methylpyrazine-M/D, 2,6-dimethylpyrazine-M/D, and 2,3-dimethylpyrazine, appeared in the squid mantles (121 °C treatment). However, the concentrations of acetone, ethanol, 1-hydroxy-2-propanone-M, ethyl formate, and dimethyl sulfide-D in the lower-temperature-treatment groups (UH, 80, 90, and 100 °C) were significantly higher than those of the 121 °C treated squid mantles (*p* < 0.05). No significant differences were observed in the concentrations of methyl acetate and 2,3-butanedione between the UH samples and the squid mantles treated with the selected temperatures (80, 90, and 100 °C), indicating that these three experimental temperatures slightly affected the concentrations of the two flavor substances. Moreover, the concentration of chemical substances related to 2-butanone, 1-hydroxy-2-propanone-M, 6-methyl-5-hepten-2-one, 3-hydroxybutan-2-one-M, 2-methyl-2-propanol, furfural, allyl sulfide, and 3-ethylpyridine exhibited a positive correlation with temperature among the squid mantles, except for the sample treated with 121 °C.

#### 3.5.3. Flavor Characterization Analysis

The detailed VOCs of all the samples are presented in Table 3. A total of 41 compounds were identified, including 33 monomers (7 alcohols, 6 aldehydes, 6 ketones, 3 esters, 4 pyrazines, 3 sulfides, 1 acid, and 3 other compounds) and 8 dimers (3-hydroxybutan-2-one-D, propanal-D, hexanal-D, 1-hydroxy-2-propanone-D, 2-methylpyrazine-D, 2,6-dimethylpyrazine-D, dimethyl sulfide-D, and acetic acid-D).

##### Alcohols

Alcohols are usually derived from chemical reactions such as lipid oxidation, decarboxylation, and the dehydrogenation of amino acids [46]. As shown in Table 3, significant differences (*p* < 0.05) were observed in total alcohol contents between the high-temperature (121 °C)-treated squid mantles and the other samples, indicating that a temperature of 121 °C facilitated the oxidation of lipids and the formation of alcohols. Liu also found that a mutton sample deep-fried at 280 °C exhibited the highest alcohol content compared to samples boiled at 100 °C and baked at 180 °C, and that the alcohols were mainly generated through the oxidation of linoleic acid degradation products [47]. Additionally, linoleic acid is the main component of fatty acids in squid mantles. This can explain why the highest alcohol content was found in the squid mantles treated at 121 °C. Overall, the content of alcohols exhibited a positive relationship with temperature, except for 2-methyl-2-propanol and 1-penten-3-ol. The contents of these two alcohols in the 121 °C treated squid mantles were significantly lower than those of the other groups (*p* < 0.05). According to the results obtained by Mariutti and Bragagnolo, 2-methyl-2-propanol and 1-penten-3-ol were derived from lipid oxidation and contributed to the flavor of mushrooms and grass [48]. Additionally, due to the high odor threshold, the effects of alcohols on the final flavor of meat products were not as significant as those of aldehydes, but the superimposition of alcohols and aromatic substances (such as aldehydes and ketones) also contributed to the flavor of meat food [49].

##### Aldehydes

Aldehydes play a key role in the flavor of aquatic products with a low odor threshold [50]. As shown in Table 3, six aldehydes, including their Ms and Ds, which are detected in meat products, were also found in the UH samples and the heating-treated squid mantles. The content of acetaldehyde dramatically decreased with the increase in temperature (6764 in the UH sample, 2269 in the 121 °C treated sample). The decrease in the acetaldehyde content could probably be attributed to the low boiling point of acetaldehyde [51]. In contrast, the content of furfural increased with increasing temperature (133.7 in the UH sample, 533.7 in the 121 °C treated sample). This phenomenon was attributable to the occurrence of the Maillard reaction, which had a positive intensity relationship with temperature and mainly contributed to the production of furfural. The content of hexanal (including its Ds and Ms) in the squid mantles exhibited a similar tendency. The hexanal content of the samples treated at 80 °C was lower than that of the UH sample. However, the hexanal content of the heating-treated samples gradually increased with the increasing processing temperature, indicating a close relationship between the production of hexanal and the heating temperature. High concentrations of hexanal are usually associated with rancid and unpleasant odors [52]. This result might explain the lowest sensory evaluation scores of the 121 °C treated squid mantles. Additionally, the ion intensity of 2-methyl propanal and 2-methylbutanal varied from 532.94 to 571.21 and from 5983.99 to 5559.58 in the UH and the 121 °C treated samples, respectively, illustrating insignificant changes in the contents of the two aldehydes in all the samples. The derivative of propanal was identified as the main volatile component in processed squid and regarded as the key contributor to the flavor of the squid product [17]. 2-methylbutanal was the most abundant aldehyde detected among the samples, especially in the lower-temperature-treated squid mantles (80, 90, and 100 °C). Therefore, 2-methylbutanal might mainly contribute to the flavor of the squid in this study.

##### Ketones

Chemical reactions, such as lipid oxidation, the Maillard reaction, and amino acid degradation, are prone to producing ketones [53]. Ketones exhibit a creamy and fruity aroma, thus positively contributing to the final flavor of the food product. As shown in Table 3, the ion intensity of 1-hydroxy-2-propanone (D) and 3-hydroxybutan-2-one (D) significantly increased in the 121 °C treated samples (3996.82 and 3583.74, respectively), indicating that the contents of these two ketones significantly increased. Similarly, the content of ketones increased with increasing heating temperatures, except for acetone. The contents of ketones (except for acetone) in the squid mantles treated at 121 °C were dramatically higher than those of aldehydes and alcohols. Cui et al. found no significant relationship between the concentration of ketones and the processing temperature [18]. Thus, considering the low content of fat in squid mantles (approximately 1.44%), the changes in ketones in this study might be attributable to the process of frying. Soybean oil is rich in unsaturated fatty acids. The introduction of external unsaturated fatty acids contributed to lipid oxidation, aggravating the production of ketones. Additionally, although the total ion intensity of ketones in the 121 °C treated squid mantles was 30309.46, which was significantly higher than that of the other samples, ketones were considered less important than aldehydes because of their high sensory threshold [54]. Additionally, chemical reactions such as lipid oxidation played a dual role (a combination of contribution of special flavor and production of potential security risks) during the heating procedure. Thus, the content of malonaldehyde (MDA, the main harmful product of lipid oxidation) in the 121 °C treated squid mantles and the UH samples was examined, and no significant differences in MDA content were observed between these two samples (*p* > 0.05).

##### Pyrazines, Sulfides, Esters, and Acids

Three pyrazines (including their Ds and Ms) were detected in the UH samples and the heating-treated samples. The ion intensity of pyrazines in the 121 °C treated squid mantles was the highest, indicating a strong, unpalatable baking taste, which was consistent with the results of the sensory evaluation [55]. Sulfides were indicative of the degradation of proteins and amino acids, which comprised sulfur-containing chemical groups. Three sulfides were found in all samples. Dimethyl sulfide is a simple sulfide-containing substance among sulfides, and it was derived from the destructed histidine resulting from heating. Three esters with low odor thresholds were detected in all samples. The intensity results of the three esters revealed that temperature affected the content of esters in the heating-treated samples, especially ethyl formate, which decreased with the increase in temperature. Acids are not usually considered a primary source of taste. The panelists did not detect any sour taste in the squid mantles in the sensory evaluation. Significantly, among the three remaining chemical substances, it was noteworthy that the intensity of trimethylamine dramatically increased, especially for the sample treated at 121 °C, which exhibited an intensity of 13,851.31. The content of trimethylamine is frequently used to evaluate the deterioration of aquatic products. However, in this study, the formation of trimethylamine was not subjected to the growth of microorganisms, which did not survive at the selected experimental temperatures. Thus, the trimethylamine in the samples was endogenous. Additionally, according to the results of Carvalho et al., trimethylamine was the main product after the thermal disintegration of TMAO, which is a natural component of squid muscle [56]. The formation of trimethylamine relied on the treatment temperature; the higher the temperature, the more trimethylamine was generated. Thus, this explained the reason for the continuous increase in trimethylamine intensity in the different temperature-treatment groups.

#### 3.5.4. Electronic Nose

Because of the advantage of collecting useful information that affected the spatial distribution of the samples, principal component analysis was used to analyze the differences among the samples [57]. The results presented in Figure 4A are consistent with the variation in volatiles obtained from HS-GC-IMS, and the total variance in the contribution of the first and second principal components was 98.419%. The variance in the contribution of the first principal component was 96.063%, indicating that the two principal components represented the main information characteristics of the heating-treated samples (Figure 4B). Additionally, there was no intersection or overlap between the different samples, indicating good discrimination among the five heating-treated samples with different temperatures. Among them, the sample treated at 121 °C was significantly different from the other samples, which indicated that a treatment temperature of 121 °C negatively impacted the flavor of the squid mantles.

## 4. Conclusions

In this study, the effect of different heating temperatures on the physicochemical and volatile flavor properties of fried Argentinian shortfin mantles was investigated. The results showed that lower treating temperatures (80 and 90 °C) improved chromatic and textural properties, along with organoleptic perception. The heating treatment at 121 °C significantly decreased the total content of free amino acids in the muscle proteins of the squid mantles. A total of 41 VOCs were identified in all the squid mantles using HS-GC-IMS, and aldehydes mainly contributed to the flavor of the squid. The results of the E-nose showed that a treatment temperature of 121 °C negatively impacted the flavor of the squid mantles. This study provided valuable insights for improving the flavor attributes of squid mantles and related cephalopod products during heating. The findings of this study could serve as a reference point for food industries in understanding the flavor profile of cooked squid and related aquatic products.

## Figures and Tables

**Figure 1 foods-13-01025-f001:**
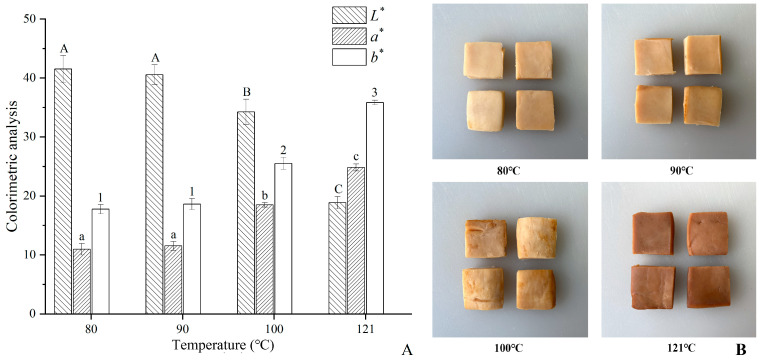
Values of *L**, *a**, *b** (**A**) and images of squid mantles (**B**) treated with different temperatures. Different letters and numbers in the same group suggest significant differences (*p* < 0.05).

**Figure 2 foods-13-01025-f002:**
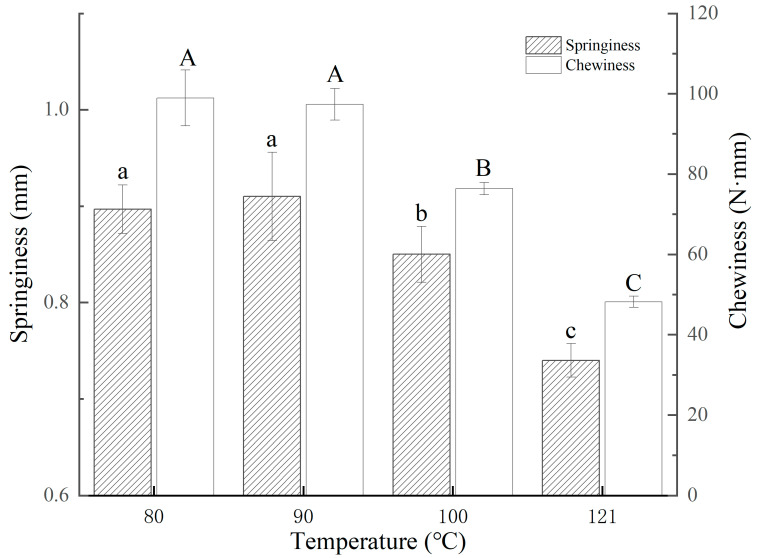
Values of springiness and chewiness of squid mantles treated with different temperatures. Different letters in the same group suggest significant differences (*p* < 0.05).

**Figure 3 foods-13-01025-f003:**
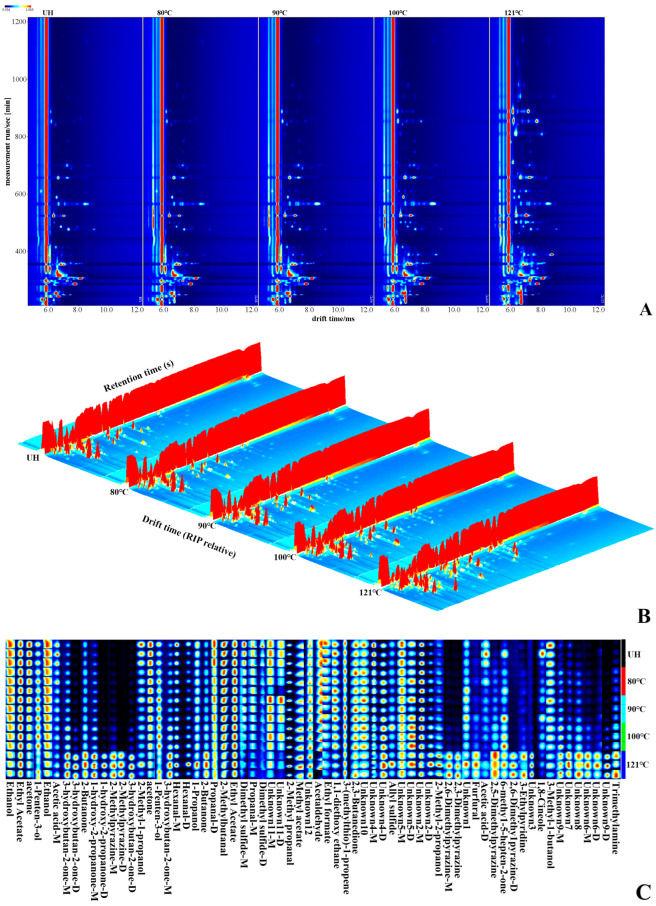
HS-GC-IMS 2D topographic plot (**A**), 3D topographic plot (**B**), and fingerprint spectra (**C**) of VOCs in squid mantles treated with different temperatures.

**Figure 4 foods-13-01025-f004:**
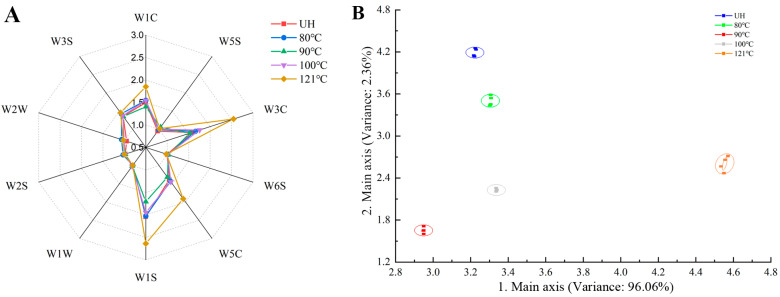
Radar image (**A**) and principal component analysis score plot (**B**) of E-nose in squid mantles treated with different temperatures.

**Table 1 foods-13-01025-t001:** Sensory evaluation of squid mantles treated with different temperatures.

Temperature (°C)	Odor	Taste	Tenderness	Juiciness	Total Score
80	6.4 ± 0.7 ^a^	6.0 ± 0.9 ^a^	5.6 ± 0.4 ^a^	5.8 ± 0.6 ^a^	6.0 ± 0.6 ^a^
90	6.6 ± 0.5 ^a^	6.0 ± 0.5 ^a^	5.4 ± 0.5 ^a^	5.8 ± 0.5 ^a^	6.1 ± 0.4 ^a^
100	5.4 ± 0.8 ^b^	5.2 ± 0.4 ^b^	4.2 ± 0.4 ^b^	4.8 ± 0.7 ^b^	5.1 ± 0.3 ^b^
121	3.5 ± 0.5 ^c^	1.9 ± 0.7 ^c^	4.4 ± 0.5 ^b^	3.4 ± 0.5 ^b^	2.8 ± 0.5 ^c^

Data are presented as the mean ± standard deviation of three replicates. Different letters in the same line indicate a significant difference (*p* < 0.05).

**Table 2 foods-13-01025-t002:** Composition and content (mg/g) of free amino acids in unheated samples and squid mantle muscle treated with different temperatures.

FAA	Threshold	Taste Characteristics	UH		80 °C		90 °C		100 °C		121 °C	
			Content	TAV	Content	TAV	Content	TAV	Content	TAV	Content	TAV
Asp	100	fresh	23.0 ± 0.8 ab	0.23	21.7 ± 0.5 b	0.22	17.0 ± 0.0 c	0.17	23.7 ± 1.3 a	0.24	17.3 ± 0.5 c	0.17
* Thr	260	sweet	20.3 ± 0.9 b	0.08	22.0 ± 0.0 a	0.08	19.3 ± 0.5 b	0.07	19.0 ± 0.8 b	0.07	14.3 ± 0.5 c	0.06
Ser	150	sweet	16.3 ± 0.9 ab	0.11	17.0 ± 0.0 a	0.11	15.3 ± 0.5 b	0.10	16.0 ± 0.8 ab	0.11	12.3 ± 0.5 c	0.08
Glu	30	fresh	246.7 ± 12.5 ab	8.22	226.7 ± 4.7 b	7.56	190.0 ± 0.0 c	6.33	260.0 ± 16.3 a	8.67	143.3 ± 4.7 d	4.78
Gly	130	sweet	21.0 ± 0.8 a	0.16	22.0 ± 0.0 a	0.17	22.3 ± 0.5 a	0.17	19.0 ± 0.8 b	0.15	15.3 ± 0.5 c	0.12
Ala	60	sweet	90.7 ± 3.4 b	1.51	100.0 ± 0.0 a	1.67	100.0 ± 0.0 a	1.67	88.7 ± 4.2 bc	1.48	83.3 ± 2.6 c	1.39
Cys	-	tasteless	1.8 ± 0.1 b	-	2.0 ± 0.1 a	-	2.0 ± 0.1 a	-	1.6 ± 0.1 c	-	1.5 ± 0.1 d	-
* Val	40	bitter	16.3 ± 0.9 ab	0.41	17.3 ± 0.5 a	0.43	15.3 ± 0.5 b	0.38	16.3 ± 0.9 ab	0.41	13.3 ± 0.5 c	0.33
* Met	30	bitter	20.3 ± 0.9 a	0.68	21.0 ± 1.4 a	0.70	21.7 ± 0.5 a	0.72	16.3 ± 0.9 b	0.54	13.7 ± 0.5 c	0.46
Iso	90	bitter	9.5 ± 0.8 a	0.11	8.4 ± 1.3 a	0.09	8.9 ± 0.3 a	0.10	10.1 ± 1.3 a	0.11	7.9 ± 0.4 a	0.09
* Leu	190	bitter	24.3 ± 2.4 ab	0.13	20.3 ± 2.9 b	0.11	22.0 ± 0.8 ab	0.12	27.3 ± 3.1 a	0.14	23.0 ± 1.4 ab	0.12
Tyr	-	bitter	10.8 ± 1.1 ab	-	8.1 ± 1.4 c	-	9.0 ± 0.6 bc	-	11.3 ± 0.9 a	-	7.6 ± 0.3 c	-
* Phe	90	bitter	2.8 ± 0.3 b	0.03	1.8 ± 0.1 c	0.02	2.2 ± 0.4 bc	0.02	3.6 ± 0.3 a	0.04	1.7 ± 0.1 c	0.02
* Lys	50	tasteless	18.3 ± 0.9 a	0.37	18.0 ± 0.0 a	0.36	15.0 ± 0.0 b	0.30	16.7 ± 1.3 ab	0.33	12.7 ± 0.5 c	0.25
His	20	bitter	13.3 ± 0.5 b	0.67	14.0 ± 0.0 a	0.70	12.0 ± 0.0 c	0.60	6.6 ± 0.3 d	0.33	2.5 ± 0.1 e	0.13
Arg	50	bitter	120.0 ± 0.0 a	2.40	113.3 ± 4.7 a	2.27	91.0 ± 0.8 b	1.82	95.0 ± 4.6 b	1.90	58.0 ± 1.6 c	1.16
Pro	300	tasteless	106.7 ± 4.7 b	0.36	110.0 ± 0.0 b	0.37	120.0 ± 0.0 a	0.40	106.0 ± 5.7 b	0.35	93.3 ± 2.6 c	0.31
TAA			762.2 ± 31.2 ^a^		743.7 ± 1.9 ^a^		683.1 ± 5.0 ^b^		737.3 ± 42.5 ^ab^		521.3 ± 15.4 ^c^	

Data are presented as the mean ± standard deviation of three replicates. Additional letters in the same row indicate a significant difference (*p* < 0.05); free amino acids with “*” are essential amino acids; TAA: total free amino acid content; TAV: taste activity value.

**Table 3 foods-13-01025-t003:** HS-GC-IMS database identification of 41 volatile compounds in squid mantles treated with different temperatures.

Volatiles	No.	Compound	CAS	Retention	Retention	Drift Time	Intensity (V)				
				Index	Times (s)	(ms)	UH	80 °C	90 °C	100 °C	121 °C
Alcohols	1	Ethanol	C64175	922.4	319.1	1.1	4738.1 ± 224.6 ^c^	5924.7 ± 92.7 ^b^	6540.8 ± 148.6 ^a^	6523.4 ± 106.1 ^a^	6568.1 ± 108.7 ^a^
	2	1-Propanol	C71238	1026	416.8	1.1	885.6 ± 194.9 ^a^	798.1 ± 28.4 ^a^	780.5 ± 40.6 ^a^	807.0 ± 73.5 ^a^	829.9 ± 47.2 ^a^
	3	2-Methyl-2-propanol	C75650	902.8	306.1	1.3	673.1 ± 52.9 ^a^	469.3 ± 16.1 ^b^	386.1 ± 31.5 ^c^	373.3 ± 26.6 ^c^	331.5 ± 23.0 ^c^
	4	2-Methyl-1-propanol	C78831	1073.5	487.4	1.2	620.1 ± 201.4 ^b^	566.1 ± 39. 4 ^b^	644.0 ± 42.7 ^b^	611.8 ± 135.1 ^b^	996.7 ± 95.8 ^a^
	5	3-Methyl-1-butanol	C123513	1179.1	702.5	1.2	434.9 ± 64.8 ^c^	573.9 ± 28.6 ^b^	593.1 ± 36.3 ^b^	624.3 ± 75.1 ^a^	677.8 ± 71.1 ^a^
	6	1-Penten-3-ol	C616251	1136.4	613.8	0.9	1104.4 ± 138.4 ^ab^	1236.5 ± 39.6 ^a^	973.4 ± 41.6 ^bc^	1146.4 ± 20.7 ^a^	827.9 ± 20.4 ^c^
	7	1,8-Cineole	C470826	1174.2	693.1	1.3	346.9 ± 65.3 ^b^	538.1 ± 122.9 ^a^	396.3 ± 79.2 ^b^	575.1 ± 226.2 ^a^	597.5 ± 331.3 ^a^
Total							8803.22 ^a^	10,106.8 ^b^	10,314.02 ^b^	10,661.9 ^b^	10,829.5 ^c^
Aldehydes	8	Acetaldehyde	C75070	753.8	223.4	0.9	6764.6 ± 179.9 ^a^	5978.5 ± 255.7 ^ab^	5985.9 ± 220.6 ^ab^	4694.1 ± 212.8 ^b^	2269.1 ± 1357.6 ^c^
	9	Propanal-M	C123386	801.8	247.3	1.0	1825.1 ± 76.9 ^a^	1631.5 ± 57.9 ^b^	1468.8 ± 39.6 ^c^	1175.3 ± 35.8 ^d^	741.4 ± 56.3 ^e^
	10	Propanal-D	C123386	801.8	247.3	1.1	5113.2 ± 234.8 ^a^	4243.6 ± 262.3 ^b^	4618.8 ± 347.7 ^ab^	4314.7 ± 112.7 ^b^	4464.7 ± 54.8 ^b^
	11	2-Methyl propanal	C78842	808	250.6	1.3	532.9 ± 43.4 ^a^	532.2 ± 56.4 ^a^	538.9 ± 65.1 ^a^	571.2 ± 23.7 ^a^	559.2 ± 62.6 ^a^
	12	2-Methylbutanal	C96173	907.8	309.4	1.4	5984.0 ± 168.0 ^a^	6173.0 ± 214.2 ^a^	6168.4 ± 233.0 ^a^	6346.3 ± 30.1 ^a^	5559.6 ± 69.7 ^b^
	13	Hexanal-M	C66251	1069.1	479.8	1.3	1502.3 ± 220.0 ^a^	1229.3 ± 82.7 ^a^	1273.6 ± 200.4 ^a^	1393.0 ± 91.7 ^a^	1557.5 ± 267.0 ^a^
	14	Hexanal-D	C66251	1070	481.2	1.6	395.2 ± 114.6 ^a^	285.2 ± 38.6 ^a^	329.2 ± 83.1 ^a^	393.7 ± 62.9 ^a^	517.7 ± 186.0 ^a^
	15	Furfural	C98011	1425.7	1383.4	1.1	133.7 ± 23.4 ^b^	140.1 ± 36.0 ^b^	184.5 ± 52.5 ^b^	203.0 ± 9.8 ^b^	533.7 ± 59.3 ^a^
Total							22,200.9 ^a^	20,193.4 ^b^	20,558.1 ^b^	19,111.9 ^c^	16,203.0 ^d^
Ketones	16	Acetone	C67641	819.4	256.7	1.1	7751.0 ± 294.0 ^a^	7328.2 ± 145.2 ^ab^	7072.2 ± 287.3 ^b^	7208.6 ± 116.2 ^ab^	6829.1 ± 368.3 ^b^
	17	1-Hydroxy-2-propanone-M	C116096	1264.2	888.2	1.0	1417.5 ± 75.0 ^d^	1647.3 ± 72.8 ^c^	1786.2 ± 96.0 ^c^	2370.7 ± 122.6 ^b^	5050.6 ± 123.6 ^a^
	18	1-Hydroxy-2-propanone-D	C116096	1265.7	891. 9	1.2	103.1 ± 10.2 ^b^	153.4 ± 12.4 ^b^	184.1 ± 20.2 ^b^	363.6 ± 51.1 ^b^	3996.8 ± 489.2 ^a^
	19	2-Butanone	C78933	893.3	300.0	1.2	1380.2 ± 94.5 ^c^	1678.0 ± 85.7 ^c^	1710.2 ± 39.1 ^c^	2152.7 ± 100.6 ^b^	4160.4 ± 370.5 ^a^
	20	3-Hydroxybutan-2-one-M	C513860	1251.5	857.8	1.0	1499.1 ± 47.6 ^d^	1780.3 ± 103.5 ^c^	1874.3 ± 73.6 ^c^	2644.5 ± 86.7 ^b^	3770.0 ± 29.4 ^a^
	21	3-Hydroxybutan-2-one-D	C513860	1250.5	855.3	1.3	182.1 ± 7.0 ^c^	263.9 ± 23.5 ^c^	304.9 ± 14.4 ^c^	732.9 ± 63.2 ^b^	3583.7 ± 193.6 ^a^
	22	2,3-Butanedione	C431038	977.3	358.3	1.2	2399.2 ± 69.4 ^a^	2674.9 ± 163.9 ^a^	2819.1 ± 162.1 ^a^	2818.1 ± 25.1 ^a^	2528.1 ± 304.3 ^a^
	23	6-Methyl-5-hepten-2-one	C110930	1300.9	982.4	1.2	205.3 ± 101.1 ^a^	238.01 ± 54.7 ^a^	282.6 ± 118.4 ^a^	367.2 ± 57.6 ^a^	390.8 ± 49.4 ^a^
Total							14,937.5 ^a^	15,764.1 ^b^	16,033.4 ^b^	18,658.4 ^c^	30,309.5 ^d^
Pyrazines	24	2-Methylpyrazine-M	C109080	1232.3	813.7	1.1	277.0 ± 11.1 ^c^	376.4 ± 34.6 ^c^	411.8 ± 49.4 ^c^	799.3 ± 14.9 ^b^	4773.7 ± 132.1 ^a^
	25	2-Methylpyrazine-D	C109080	1231.1	810.9	1.4	40.5 ± 4.1 ^b^	38.1 ± 3.5 ^b^	36.2 ± 1.4 ^b^	45.9 ± 4.8 ^b^	1004.0 ± 74.0 ^a^
	26	2,3-Dimethylpyrazine	C5910894	1301.1	982.9	1.1	75.6 ± 5.5 ^c^	92.6 ± 9.1 ^bc^	103.4 ± 9.9 ^bc^	139.1 ± 0.9 ^b^	1003.6 ± 50.9 ^a^
	27	2,5-Dimethylpyrazine	C123320	1265.1	890.4	1.1	85.3 ± 18.9 ^d^	156.5 ± 6.9 ^c^	191.3 ± 14.7 ^c^	287.6 ± 16.0 ^b^	532.2 ± 34.8 ^a^
	28	2,6-Dimethylpyrazine-M	C108509	1286.8	944.9	1.1	69.9 ± 5.1 ^c^	99.9 ± 8.2 ^c^	104.1 ± 13.7 ^c^	192.9 ± 8.2 ^b^	1420.8 ± 60.2 ^a^
	29	2,6-Dimethylpyrazine-D	C108509	1285.8	942.4	1.5	69.9 ± 5.1 ^c^	99.9 ± 8.2 ^c^	104.1 ± 13.7 ^c^	192.9 ± 8.2 ^b^	1420.8 ± 60.2 ^a^
Total							593.8 ^a^	811.7 ^b^	893.0 ^b^	1509.7 ^c^	9021.6 ^d^
Sulfides	30	Dimethyl sulfide-M	C75183	776.9	234.6	0.9	5115.3 ± 168.8 ^a^	5142.6 ± 306.2 ^a^	5565.6 ± 206.2 ^a^	5193.1 ± 178.1 ^a^	3653.0 ± 550.9 ^b^
	31	Dimethyl sulfide-D	C75183	772.7	232.5	1.1	1073.0 ± 178.8 ^a^	715.4 ± 93.9 ^b^	726.4 ± 103.2 ^b^	619.2 ± 25.6 ^b^	467.1 ± 123.1 ^b^
	32	3-(Methylthio)-1-propene	C10152768	958.1	344.1	1.0	4164.3 ± 99.1 ^a^	4127.5 ± 243.2 ^a^	4289.8 ± 233.4 ^a^	4161.0 ± 288.6 ^a^	3930.9 ± 158.9 ^a^
	33	Allyl sulfide	C592881	1126.4	591.7	1.1	1127.4 ± 107.9 ^a^	1087.4 ± 155.5 ^a^	1210.9 ± 117.4 ^a^	1471.0 ± 386.1 ^a^	1602.564.4 ^a^
Total							11,480.4 ^a^	11,072.8 ^a^	11,792.8 ^a^	11,444.4 ^a^	9653.8 ^b^
Esters	34	Methyl acetate	C79209	810.7	251.9	1.2	452.8 ± 28.9 ^a^	457.4 ± 24.4 ^a^	506.28.0 ^a^	514.4 ± 8.8 ^a^	476.9 ± 31.4 ^a^
	35	Ethyl Acetate	C141786	875.3	288.9	1.3	7731.0 ± 84.2 ^a^	7677.3 ± 401.3 ^b^	7469.3 ± 306.1 ^b^	7761.5 ± 92.7 ^a^	6994.3 ± 434.0 ^b^
	36	Ethyl formate	C109944	807.1	250.1	1.2	251.5 ± 19.2 ^a^	208.5 ± 17.5 ^ab^	207.7 ± 28.7 ^ab^	188.3 ± 4.1 ^bc^	160.3 ± 22.3 ^c^
Total							8435.3 ^a^	8343.3 ^a^	8183.1 ^b^	8464.2 ^a^	7631.6 ^c^
Acids	37	Acetic acid-M	C64197	1435.5	1420.9	1.0	6448.6 ± 347.6 ^a^	5129.2 ± 587.9 ^b^	4879.4 ± 74.1 ^b^	4610.7 ± 306.0 ^b^	4762.4 ± 309.1 ^b^
	38	Acetic acid-D	C64197	1432.9	1411.1	1.2	367.4 ± 38.5 ^a^	240.7 ± 64.2 ^b^	219.6 ± 15.8 ^b^	204.2 ± 39.8 ^b^	166.8 ± 34.8 ^b^
Total							6816.0 ^a^	5369.9 ^b^	5099.0 ^c^	4814.9 ^d^	4929.2 ^d^
Others	39	1,1-Diethoxy ethane	C105577	885.8	295.3	1.0	1798.8 ± 160.3 ^a^	1794.4 ± 121.5 ^a^	1960.6 ± 31.1 ^a^	1740.8 ± 95.5 ^a^	1241.1 ± 39.9 ^b^
	40	Trimethylamine	C75503	765.4	228.9	1.1	1092.9 ± 303.5 ^c^	3104.1 ± 276.7 ^c^	3928.9 ± 430.0 ^bc^	7237.9 ± 592.4 ^b^	13,851.3 ± 3537.03 ^a^
	41	3-Ethylpyridine	C536787	1327.5	1056.6	1.1	81.2 ± 9.9 ^b^	130.4 ± 10.2 ^b^	126.3 ± 8.9 ^b^	153.6 ± 17.5 ^b^	344.3 ± 82.6 ^a^

Data are presented as the mean ± standard deviation of three replicates. Different letters in the same row indicate a significant difference (*p* < 0.05).

## Data Availability

The data presented in this study are available on request from the corresponding author. The data are not publicly available due to privacy restrictions.

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
