# Peer review of "Effects of Heating Treatment on the Physicochemical and Volatile Flavor Properties of Argentinian Shortfin Squid *(Illex argentinus*)"

_foods, 2024, doi:10.3390/foods13071025_

Round 1
Reviewer 1 Report
Comments and Suggestions for Authors
The manuscript provides important results and conclusions regarding the effect of different heating temperatures (80, 90, 100, and 121°C) on the physicochemical and volatile flavor properties of fried mantles (Argentinian shortfin). The scope of this paper fits with that of the journal Foods.
The manuscript presents an interesting subject which would be interesting for many readers. Therefore, in my opinion, it presents a good degree of originality and I think that it is suitable for the journal.
Many different analyses have been carried out, which greatly enhances the scientific value of this study, it is easy to read and understand, however, there are some points which the authors should correct as I recommend a major revision.
The following are my remarks in detail
Page 2- Line 35. Delete ‘Among the species’
Page 2- Line 36. (Illex. argentinus) Delete the period
Page 2- Line 59. ….derived from these chemical reactions mentioned above…. Rewrite ‘derived from the above mentioned chemical reactions’
Page 2- Line 74. ……traditional processing methods and the effect of adding… traditional processing methods and on the effect of adding….
Page 2- Line 76. ….. al., 2019; Sanz et al., 2004), replace the comma with a period and start a new sentence.
Page 2- Line 77. ……of squid products. Add references
Page 3- Colorimetric analysis…. explains the method better with the meanings of the letters L, a and b.
Page 3- Sensory evaluation. Did the assessors remain without eating and smoking for at least two hours before analysis? Did they only drink water during the tasting? Add these information.
Page 6- Line 225…unheated (UH) samples (the squid mantles that were similarly treated like other samples but without heating treatment)… it should be put in the materials and methods.
Page 6- Line 227-230. However, no significant differences were observed in amino acid composition or content between the UH sample and the squid mantles treated at 80 and 90°C (P > 0.05), indicating that the heating treatment with the lower temperature (80 and 90°C) has no significant effect on the squid mantle muscle protein.
This sentence is redundant…rewrite…such as … the heating treatment with the lower temperatures (80 and 90°C) has no significant effect on the amino acid composition or content of the squid mantle muscle protein..
Page 6- Line 237-238. The samples were treated with 80°C (743.67 mg/g) and 90°C (737.27 mg/g) (P < 0.05), showing a negative correlation between the temperature and total amino acid content.
The samples treated with 80°C (743.67 mg/g) and 90°C (737.27 mg/g) (P < 0.05) showed a negative correlation between the temperature and total amino acid content.
Page 6- Line 240- 242 According to the results in Table 2, the amino acids with the TAV value higher than 1 of all samples were glutamic acid, alanine, and arginine, and the amino acid composition was independent of processing temperature. Rewrite this sentence more clearly.
Page 6- Line 243 …in the sample (121°C)…rewrite… in the 121°C-treated sample
Page 6- Line 246. Meanwhile, the arginine content of the sample (121°C treatment) was ….
Meanwhile, the arginine content of the 121°C sample was….
Page 7- Line 261-262 .Additionally, all four samples exhibited a similar trend in chewiness values to that of springiness values.
Rewrite… Additionally, all samples showed a similar trend in chewiness and springiness values.
Page 7- Line 264- (treated at 80 and 90°C)… Delete the brackets.
Page 9- Line 298- 299. …especially for the squid mantles treated at 121°C, significantly changed owing to the height and the brightness of spots.
….. especially for those treated at 121°C, significantly changed as shown by the height and brightness of the spots.
Page 11- Line 334. (including seven alcohols,…Delete including….
Page 14- Line 346. indicating that that…Delete that
Page 14- Line 366-367. The content of acetaldehyde dramatically decreased with the increase in temperature. Put the range, also in the brackets.
Page 14- Line 368-369. In contrast, the content of furfural increased with increasing temperature. Also in this case put the range to make it more immediate to the reader.
Page 14- Line 378-379. Additionally, the ion intensity of 2-methyl propanal and 2-methylbutanal varied from 532.94–571.21 and 5983.99–5559.58, respectively.
Explains better such as…. the ion intensity of 2-methyl propanal and 2-methylbutanal varied from 532.94 to 571.21 and from 5983.99 to 5559.58 in UH and 121°C treated sample, respectively.
Page 14- Line 390-391. Explains better such as ….As shown in Table. 3, ion intensity of 1-hydroxy-2-propanone (D) and 3-hydroxybutan-2-one (D) significantly increased in 121°C treated samples (3996.82 and 3583.74, respectively).
Comments on the Quality of English Language
Could be improved
Author Response
1. Comments 1: Page 2- Line 35. Delete ‘Among the species’
--Response: ” Among the species” were deleted.
2. Comments 2: Page 2- Line 36. (Illex. argentinus) Delete the period
--Response: All the period between the “Illex. argentinus” were deleted. Awfully sorry for this mistake, and thank reviewer for your scientificity and preciseness
3. Comments 3: Page 2- Line 59. ….derived from these chemical reactions mentioned above…. Rewrite ‘derived from the above mentioned chemical reactions’
--Response: It was revised.
4. Comments 4:Page 2- Line 74. ……traditional processing methods and the effect of adding… traditional processing methods and on the effect of adding….
--Response: It was revised.
5. Comments 5:Page 2- Line 76. ….. al., 2019; Sanz et al., 2004), replace the comma with a period and start a new sentence.
--Response: It was revised.
6. Comments 6: Page 2- Line 77. ……of squid products. Add references
--Response: Related reference was added.
7. Comments 7: Page 3- Colorimetric analysis…. explains the method better with the meanings of the letters L, a and b.
--Response: The meanings of the letters L, a and b were explained.
8. Comments 8: Page 3- Sensory evaluation. Did the assessors remain without eating and smoking for at least two hours before analysis? Did they only drink water during the tasting? Add these information.
--Response: All the necessary information about the demands for the assessors were added.
9. Comments 9: Page 6- Line 225…unheated (UH) samples (the squid mantles that were similarly treated like other samples but without heating treatment)… it should be put in the materials and methods.
--Response: It was revised.
10. Comments 10: Page 6- Line 227-230. However, no significant differences were observed in amino acid composition or content between the UH sample and the squid mantles treated at 80 and 90°C (P > 0.05), indicating that the heating treatment with the lower temperature (80 and 90°C) has no significant effect on the squid mantle muscle protein.
This sentence is redundant…rewrite…such as … the heating treatment with the lower temperatures (80 and 90°C) has no significant effect on the amino acid composition or content of the squid mantle muscle protein..
--Response: It was revised according to the comment of reviewer.
11. Comments 11: Page 6- Line 237-238. The samples were treated with 80°C (743.67 mg/g) and 90°C (737.27 mg/g) (P < 0.05), showing a negative correlation between the temperature and total amino acid content.
The samples treated with 80°C (743.67 mg/g) and 90°C (737.27 mg/g) (P < 0.05) showed a negative correlation between the temperature and total amino acid content.
--Response: It was revised.
12. Comments 12: Page 6- Line 240- 242 According to the results in Table 2, the amino acids with the TAV value higher than 1 of all samples were glutamic acid, alanine, and arginine, and the amino acid composition was independent of processing temperature. Rewrite this sentence more clearly.
--Response: It was revised.
13. Comments 13: Page 6- Line 243 …in the sample (121°C)…rewrite… in the 121°C-treated sample
--Response: It was revised.
14. Comments 14: Page 6- Line 246. Meanwhile, the arginine content of the sample (121°C treatment) was ….
Meanwhile, the arginine content of the 121°C sample was….
--Response: It was revised.
15. Comments 15: Page 7- Line 261-262 .Additionally, all four samples exhibited a similar trend in chewiness values to that of springiness values.
Rewrite… Additionally, all samples showed a similar trend in chewiness and springiness values. --Response: It was revised.
16. Comments 16: Page 7- Line 264- (treated at 80 and 90°C)… Delete the brackets.
--Response: The brackets were deleted.
17. Comments 17: Page 9- Line 298- 299. …especially for the squid mantles treated at 121°C, significantly changed owing to the height and the brightness of spots.
….. especially for those treated at 121°C, significantly changed as shown by the height and brightness of the spots.
--Response: It was revised.
18. Comments 18: Page 11- Line 334. (including seven alcohols,…Delete including….
--Response: It was revised.
19. Comments 19: Page 14- Line 346. indicating that that…Delete that
--Response: It was revised.
20. Comments 20: Page 14- Line 366-367. The content of acetaldehyde dramatically decreased with the increase in temperature. Put the range, also in the brackets.
--Response: It was revised.
21. Comments 21: Page 14- Line 368-369. In contrast, the content of furfural increased with increasing temperature. Also in this case put the range to make it more immediate to the reader.
--Response: It was revised.
22. Comments 22: Page 14- Line 378-379. Additionally, the ion intensity of 2-methyl propanal and 2-methylbutanal varied from 532.94–571.21 and 5983.99–5559.58, respectively.
Explains better such as…. the ion intensity of 2-methyl propanal and 2-methylbutanal varied from 532.94 to 571.21 and from 5983.99 to 5559.58 in UH and 121°C treated sample, respectively.
--Response: It was revised.
23. Comments 23: Page 14- Line 390-391. Explains better such as ….As shown in Table. 3, ion intensity of 1-hydroxy-2-propanone (D) and 3-hydroxybutan-2-one (D) significantly increased in 121°C treated samples (3996.82 and 3583.74, respectively).
--Response: It was revised.
Thank you very much for the considerable concerns about our manuscript. We don’t know whether the revised manuscript has reached to the standard. If there are still some mistakes in the manuscript, please give us another chance to improve the manuscript, and hope that the corrections will meet with approval.
Sincerely,
Jiancong Huo (corresponding author)

Reviewer 2 Report
Comments and Suggestions for Authors
Comments to the Authors
The manuscript entitled “Effects of heating treatment on the physicochemical and volatile flavor properties of Argentinian shortfin squid (Illex. argentinus)” submitted by Li et al. evaluated the effect of different processing temperatures on physicochemical and volatile flavor properties of squid mantle. The results of this research are significant because they contribute to the understanding of the quality and flavor profile of squid processed under different temperatures. This information is important because is focused on supporting a better and more efficient commercialization of the products.
The manuscript is well-written and logistically organized, the experiment is well-designed, the discussion is consistent, and the conclusions are interesting. Some comments, observations, and suggestions are described below to improve the manuscript by the authors before its further consideration.
Abstract: Please, include p-values to denote differences (e.g. at L22, L26).
Entire manuscript: It is suggested to use the numbered style for references in the manuscripts.
L81-83. It is suggested to delete this paragraph and use it in the conclusion.
L93. Was a fresh sample used? please specify.
L122. This information belongs to a colorimeter. Please include the appropriate information regarding the equipment used.
L123. TPA abbreviation is unnecessary. It is suggested to delete it.
L158. It is recommended to standardize the nomenclature of p-value (p-values: lowercase, p in italic). Please check the entire manuscript.
L194-195. Fig 1A: It is suggested to include the complete name of the x-axis (Temperature)
L198. It is suggested to include a comparison with the published information.
L220-222. Table 1: “Data were represented as a mean ± standard deviation of three replicates. The different letters in the same line indicated a significant difference (p < 0.05)” It is suggested to relocate this part as a footer at the end of the table.
L222-223. Tenderness and total score are not specified in MM.
L222-223. It is suggested to use only one decimal point.
L252-254. “Data were represented as mean ± standard deviation of three replicates. The additional letters in the same row indicated a significant difference (p < 0.05); Free amino acids with "*" were essential amino acids; TAA: total free amino acid content” It is suggested to relocate this part as a footer at the end of the table. Define TAV and FAA abbreviations.
L254-255. It is suggested to use only one decimal point.
L339-340. “Data were represented as mean ± standard deviation of three replicates. The different letters in the same row indicated a significant difference (p < 0.05)” It is suggested to relocate this part as a footer at the end of the table.
L340-341. It is suggested to use only one decimal point.
L458. The correct abbreviation is HS-GC-IMS

Author Response
- Comments 1: Abstract: Please, include p-values to denote differences (e.g. at L22, L26).
--Response: p-values were added in the abstract to denote differences.
- Comments 2: L81-83. It is suggested to delete this paragraph and use it in the conclusion.
--Response: This paragraph was deleted and used in the conclusion.
- Comments 3: Was a fresh sample used? please specify.
--Response: The condition of sample was specified clearly.
- Comments 4:This information belongs to a colorimeter. Please include the appropriate information regarding the equipment used.
--Response: Awfully sorry for this mistake. The accurate information of the equipment was added.
- Comments 5:TPA abbreviation is unnecessary. It is suggested to delete it.
--Response: It was revised.
- Comments 6: It is recommended to standardize the nomenclature of p-value (p-values: lowercase, p in italic). Please check the entire manuscript.
--Response: It was revised.
- Comments 7: Fig 1A: It is suggested to include the complete name of the x-axis (Temperature)
--Response: It was revised.
- Comments 8: It is suggested to include a comparison with the published information.
--Response: It was revised.
- Comments 9: L220-222. Table 1: “Data were represented as a mean ± standard deviation of three replicates. The different letters in the same line indicated a significant difference (p < 0.05)” It is suggested to relocate this part as a footer at the end of the table.
--Response: It was revised.
- Comments 10: L222-223. Tenderness and total score are not specified in MM.
--Response: It was revised.
- Comments 11: L222-223. It is suggested to use only one decimal point.
--Response: It was revised.
- Comments 12: L252-254. “Data were represented as mean ± standard deviation of three replicates. The additional letters in the same row indicated a significant difference (p < 0.05); Free amino acids with "*" were essential amino acids; TAA: total free amino acid content” It is suggested to relocate this part as a footer at the end of the table. Define TAV and FAA abbreviations.
--Response: It was revised.
- Comments 13: L254-255. It is suggested to use only one decimal point.
--Response: It was revised.
- Comments 14: L339-340. “Data were represented as mean ± standard deviation of three replicates. The different letters in the same row indicated a significant difference (p < 0.05)” It is suggested to relocate this part as a footer at the end of the table.
--Response: It was revised.
- Comments 15: L340-341. It is suggested to use only one decimal point.
--Response: It was revised.
- Comments 16: L458. The correct abbreviation is HS-GC-IMS
--Response: It was revised.
Thank you very much for the considerable concerns about our manuscript. We don’t know whether the revised manuscript has reached to the standard. If there are still some mistakes in the manuscript, please give us another chance to improve the manuscript, and hope that the corrections will meet with approval.
Sincerely,
Jiancong Huo (corresponding author)

Reviewer 3 Report
Comments and Suggestions for Authors
The article titled " Effects of heating treatment on the physicochemical and volatile flavor properties of Argentinian shortfin squid (Illex. argentinus)" authored by Li et al., is an original contribution that provides information on the effect of temperature on physicochemical and volatile flavor properties of fried squid mantles. The article is well written, and the results are appropriately presented and discussed. Additionally, the bibliography is extensive and up-to-date.
Therefore, I believe this article should be considered after addressing the following minor revisions:
Line 21: "improved chromatic and textural properties simultaneously with organoleptic perception." instead of "improved chromatic/textural properties and organoleptic perception."
Line 41: "via physicochemical reactions" instead of "via a physicochemical reaction," as multiple reactions occur.
Lines 54-59: The information about the influence of temperature and cooking time on the formation of volatile compounds is repetitive. Synthesize the information to avoid repetition.
Line 64: "The final product" instead of "the product."
Line 130 and 144: Add a period at the end of the sentence.
Line 197: Keep the nomenclature consistent with the manuscript and replace "p < 0.05" with "P < 0.05". Please apply this to the Tables as well.
Line 252 (Table 2): Although the meaning of the acronym "UH" has already been indicated in the body of the text, this is the first time it is mentioned in a table. Please replace "UH" with "unheated (UH)" for better understanding. Also, specify the meaning of "TAV."
Lines 227-230: This statement is inaccurate. The table demonstrates significant effects on certain amino acids (e.g., Ala, Cys, His, etc.) between unheated (UH) samples and those treated at 80°C and 90°C. Even the total amino acid content (TAA) is significantly lower in samples heated at 90°C compared to unheated samples. Reformulate this part of the results and discussion in accordance with the obtained results.
Line 241: Provide an explanation of how TAV is calculated before discussing it.
Line 564: "mj"? Usually, the units for chewiness are "N·mm."
Lines 345-347: The table does not display differences (P < 0.05) in the total alcohol content (no letter appears in the total content line for any compound group). Please correct this omission error.
Line 356: Remove the comma after "Baragagnolo." It is unnecessary.
Lines 418-419: "The variation in the histidine content (Table 2) explained the reason for the intensity decrease in dimethyl sulfide (Table 3)." This statement is not necessarily correct, as dimethyl sulfide could have been produced from the degradation of histidine.
Thus, a higher content of dimethyl sulfide could be related to a lower content of histidine due to its degradation. Therefore, your statement may be incorrect. Consider removing it.
Line 458: "Microorganisms" instead of "organism."
Line 453: Same revision as in the abstract, "improved chromatic and textural properties simultaneously with organoleptic perception." instead of "improved chromatic/textural properties and organoleptic perception."
Author Response
- Comments 1: Line 21: "improved chromatic and textural properties simultaneously with organoleptic perception." instead of "improved chromatic/textural properties and organoleptic perception."
--Response: It was revised according to the comment.
- Comments 2: Line 41: "via physicochemical reactions" instead of "via a physicochemical reaction," as multiple reactions occur.
--Response: It was revised according to the comment.
- Comments 3: Lines 54-59: The information about the influence of temperature and cooking time on the formation of volatile compounds is repetitive. Synthesize the information to avoid repetition.
--Response: It was revised.
- Comments 4:Line 64: "The final product" instead of "the product."
--Response: The word “final” was added.
- Comments 5:Line 130 and 144: Add a period at the end of the sentence.
--Response: A period was added at the end of this sentence.
- Comments 6: Line 197: Keep the nomenclature consistent with the manuscript and replace "p < 0.05" with "P < 0.05". Please apply this to the Tables as well.
--Response: It was revised.
- Comments 7: Page 3- Colorimetric analysis…. explains the method better with the meanings of the letters L, a and b.
--Response: The meanings of the letters L, a and b were explained.
- Comments 8: Line 252 (Table 2): Although the meaning of the acronym "UH" has already been indicated in the body of the text, this is the first time it is mentioned in a table. Please replace "UH" with "unheated (UH)" for better understanding. Also, specify the meaning of "TAV."
--Response: It was revised. The meaning of TAV was specified.
- Comments 9: Lines 227-230: This statement is inaccurate. The table demonstrates significant effects on certain amino acids (e.g., Ala, Cys, His, etc.) between unheated (UH) samples and those treated at 80°C and 90°C. Even the total amino acid content (TAA) is significantly lower in samples heated at 90°C compared to unheated samples. Reformulate this part of the results and discussion in accordance with the obtained results.
--Response: It was revised, and unnecessary discussion was deleted.
- Comments 10: Line 241: Provide an explanation of how TAV is calculated before discussing it.
--Response: The calculation of TAV was added.
- Comments 11: Line 564: "mj"? Usually, the units for chewiness are "N·mm."
--Response: It was revised.
- Comments 12: Lines 345-347: The table does not display differences (P < 0.05) in the total alcohol content (no letter appears in the total content line for any compound group). Please correct this omission error.
--Response: It was revised.
- Comments 13: Line 356: Remove the comma after "Baragagnolo." It is unnecessary.
--Response: This unnecessary comma was deleted.
- Comments 14: Lines 418-419: "The variation in the histidine content (Table 2) explained the reason for the intensity decrease in dimethyl sulfide (Table 3)." This statement is not necessarily correct, as dimethyl sulfide could have been produced from the degradation of histidine. Thus, a higher content of dimethyl sulfide could be related to a lower content of histidine due to its degradation. Therefore, your statement may be incorrect. Consider removing it.
--Response: It was revised according to the comment.
- Comments 15: Line 458: "Microorganisms" instead of "organism."
--Response: It was revised. Thank reviewer for your scientificity and preciseness.
- Comments 16: Line 453: Same revision as in the abstract, "improved chromatic and textural properties simultaneously with organoleptic perception." instead of "improved chromatic/textural properties and organoleptic perception."
--Response: It was revised.
Thank you very much for the considerable concerns about our manuscript. We don’t know whether the revised manuscript has reached to the standard. If there are still some mistakes in the manuscript, please give us another chance to improve the manuscript, and hope that the corrections will meet with approval.
Sincerely,
Jiancong Huo (corresponding author)
